# Importance of Antioxidant Supplementation during In Vitro Maturation of Mammalian Oocytes

**DOI:** 10.3390/vetsci9080439

**Published:** 2022-08-18

**Authors:** Shimaa I. Rakha, Mohammed A. Elmetwally, Hossam El-Sheikh Ali, Ahmed Balboula, Abdelmonem Montaser Mahmoud, Samy M. Zaabel

**Affiliations:** 1Department of Theriogenology, Faculty of Veterinary Medicine, Mansoura University, Mansoura 35516, Egypt; 2Reproductive Biotechnology Research Laboratory, College of Veterinary Medicine, Mansoura University, Mansoura 35516, Egypt; 3Animal Sciences Research Center, University of Missouri, Columbia, MO 65211, USA

**Keywords:** antioxidants, IVEP, IVM, oocyte, oxidative stress, ROS

## Abstract

**Simple Summary:**

In vitro embryo production (IVEP) is a technology that is widely used in the field of reproductive biology. The first and most important step of IVEP is termed in vitro maturation or IVM, in which the oocyte is allowed to mature in a synthetic medium under controlled laboratory settings. The quality of in vitro matured oocytes is still low compared to that of those matured inside an animal’s body. One of the key elements that was found to affect oocyte quality is the accumulation of large amounts of reactive oxygen species (ROS) within oocytes during IVM. The addition of antioxidants to oocyte IVM media is one of the most effective methods for preventing ROS accumulation. In this article, we highlight the latest events showing how ROS harms mammalian oocytes. We also discuss the possible impacts of antioxidant supplementation on the IVM rate and oocyte quality.

**Abstract:**

The in vitro embryo production (IVEP) technique is widely used in the field of reproductive biology. In vitro maturation (IVM) is the first and most critical step of IVEP, during which, the oocyte is matured in an artificial maturation medium under strict laboratory conditions. Despite all of the progress in the field of IVEP, the quality of in vitro matured oocytes remains inferior to that of those matured in vivo. The accumulation of substantial amounts of reactive oxygen species (ROS) within oocytes during IVM has been regarded as one of the main factors altering oocyte quality. One of the most promising approaches to overcome ROS accumulation within oocytes is the supplementation of oocyte IVM medium with antioxidants. In this article, we discuss recent advancements depicting the adverse effects of ROS on mammalian oocytes. We also discuss the potential use of antioxidants and their effect on both oocyte quality and IVM rate.

## 1. Introduction

The in vitro embryo production (IVEP) technique is frequently employed to overcome problems associated with infertility in different mammalian species [1,2]. IVEP has been identified as a key technique for the production of a large number of offspring from animals with superior genetic traits [3].

The in vitro maturation (IVM) of oocytes is the first and most crucial phase in the IVEP process, during which, oocytes gain the potential to sustain continued embryonic development [4]. As a result, identifying the appropriate IVM environment is essential for effective IVEP procedures [4]. 

The success of IVM is dependent on various factors including the quality of collected oocytes and the setup of culture conditions [5]. Antioxidants supplied periodically from fluids of the female reproductive organs act to minimize the damage induced by reactive oxygen species, ROS, natural by-products of oocyte metabolism [6]. One of the challenges that oocytes face during IVM is oxidative stress caused by the increased generation of ROS by oocytes in the IVM media [7]. Excessive ROS production could ultimately result in oocyte death and embryonic loss [8,9].

Other forms of stress that oocytes may face include high ambient temperature [10], increased mother age [11], and fertilization delays caused by the failure of the sperm to reach and penetrate the oocyte membranes [12]. All these stress types represent a potential risk for female infertility since they predispose oocytes to oxidative damage and are associated with a series of physiological, metabolic, cellular, and molecular alterations in the reproductive tract [13].

Antioxidant supplementation during IVM has been suggested to protect oocytes from the injurious effects of oxidative stress via maintaining physiological levels of ROS [14,15]. In this article, current understandings of the sources of ROS during IVM of mammalian oocytes, their deleterious effects on oocyte biology, and the promising role of exogenously supplemented antioxidants are presented and discussed.

## 2. In Vitro Maturation of Mammalian Oocytes

Oocyte maturation involves a series of complex and diverse events of nuclear and cytoplasmic modifications that provide oocytes the inherent potential to promote embryo development and activate the embryonic genome [16]. Nuclear and cytoplasmic maturations are two components of oocyte maturation [17]. Cumulus expansion is also regarded as a sign of oocyte maturation [18].

Cytoplasmic maturation involves events related to the cytoplasmic capacitation of the oocyte [19]. It entails the accumulation of mRNA, proteins, and substrates necessary for fertilization and subsequent development [20]. Through the regulation of protein and ATP synthesis as well as chromosome segregation, cell organelles including mitochondria, endoplasmic reticulum (ER), and microtubules play key roles in both cytoplasmic and nuclear maturation [16,21]. 

The nuclear maturation of oocytes implies the resumption of meiosis and progression to the metaphase II stage. The mammalian oocyte passes through two successive cell divisions during its maturation process [22]. First, the oocyte is arrested at the diplotene stage. At this stage, the oocyte appears to contain a large nucleus, the germinal vesicle (GV). Germinal vesicle breakdown (GVBD) involves chromatin condensation and the disintegration of the nuclear membrane [23]. Following GVBD, the oocyte resumes meiosis and enters the metaphase I (MI) [24]. The first meiosis ends with the formation of a haploid oocyte and the extrusion of the first polar body. The former is kept at metaphase II (MII) until fertilization [25]. Inaccuracies of these meiotic events might prevent oocytes from reaching their proper maturation [26].

The process of oocyte maturation is a complex process that requires coordinated interactions between the oocyte and its surrounding cumulus cells as well as between oocyte cytoplasmic and nuclear compartments [27]. For instance, mixing of the GV contents during its breakdown with the cytoplasm has been assumed to activate critical aspects of cytoplasmic maturation and subsequent developmental competence [28]. In certain species, including equines and humans, the process of cytoplasmic maturation has been found to be deficient in several aspects, especially those related to microtubule patterning and cell division, with negative outcomes on rates of embryo formation [29,30]. Several approaches have been investigated to improve IVM in the aforementioned species by creating suitable conditions for better cytoplasmic maturation. In equines, the treatment of oocytes with the meiosis-inhibiting factor Roscovitine for a period of time prior to IVM improved microfilaments’ organization in matured oocytes [31]. Recently, co-culturing equine COCs with granulosa cells from small follicles (<15 mm) has been found to significantly improve the cytoplasmic maturation of oocytes on the basis of cortical granules’ distribution and mitochondrial function [32].

## 3. Normal Production and Elimination of Reactive Oxygen Species (ROS) by Mammalian Oocytes

ROS are oxygen-containing molecules produced as a by-product of cellular respiration and metabolism [33]. They primarily consist of superoxide anions (O_2_^−^), hydrogen peroxide (H_2_O_2_), and hydroxyl radicals (OH) [33]. The oxidation of unsaturated fatty acids, also known as lipid peroxidation, is one of the most severe outcomes of free radical attack. One of the end products of lipid peroxidation is malondialdehyde (MDA). It can be used as a cumulative marker of lipid peroxidation because it is a stable end product [34]. In oocytes, MDA is thus regarded as an indicator of oxidative stress [35].

Physiological levels of ROS have been found to be beneficial to gamete function and development [36]. However, the excessive production of ROS at levels that exceed oocyte total antioxidant capacity could result in a state of oxidative stress (OS) [13]. OS is characterized by a wide range of cellular destruction including DNA damage, arrested growth and reduced oocyte quality [37].

Intracellular ROS are repeatedly removed by a group of intracellular antioxidants. These antioxidants include an extensive set of cooperating enzymatic and non-enzymatic factors [38]. These factors are abundantly present within the follicular fluid and act to protect oocytes from harmful effects [39]. Superoxide dismutases (SODs), catalase (CAT) and glutathione peroxidase (GPx) are examples of enzymatic antioxidants. SODs convert O_2_^−^ into H_2_O_2_ via its mitochondrial, manganese-dependent superoxide dismutase (Mn-SOD) and cytoplasmic, copper, zinc superoxide dismutase (Cu/ZnSOD), while CAT and GPx mediate the breakdown of H_2_O_2_ into water and oxygen [40,41,42].

Non-enzymatic antioxidants include thiols [43], ascorbic acid, alpha-tocopherol [44,45], melatonin [46], l-carnitine [47,48], and lycopene [49]. They prevent oxidative damage by interrupting free radical chain reactions [14].

## 4. The Alteration in Oxidative Status of Mammalian Oocytes during Their IVM

The in vitro handling of gametes and embryos during ARTs is associated with the production of large amounts of ROS that exceed the typical antioxidant capacity of the cell [50]. Organelle failure, spindle abnormalities, DNA fragmentation, and apoptosis are all among the detrimental effects of high ROS on IVM oocytes [50]. 

IVM involves the culture of cumulus–oocyte complexes (COCs) in a synthetic medium. The media used for IVM usually contain fewer antioxidant enzymes than the in vivo milieu of follicular and oviduct fluids provided by the mother [6]. During IVM, OS is caused by an imbalance between ROS production and clearance induced by a shortage of maternal antioxidants. Such an alteration causes oocyte maturation and subsequent embryonic development to be disrupted [51]. 

The incubation of oocytes in high oxygen concentrations [52] and their exposure to visible light [53], pollutants [54], and certain components of IVM [50] all promote ROS generation during the in vitro maturation of mammalian oocytes. Increased ROS levels disrupted maturation-promoting factors and triggered programmed cell death in oocytes from several mammal species [9,55]. 

The lipids in the membranes of in vitro generated embryos are damaged by ROS [56]. Non-competent embryos had higher ROS levels than competent embryos, which was linked to mitochondrial damage [57]. In bovines, the presence of free radicals during IVEP reduced the blastocyst rate in a dose-dependent manner [58]. Even when the damage was not fatal, it had a detrimental impact on cellular development, metabolic activity, and embryo viability [59]. 

Oocytes are exposed to OS during IVEP not only as a result of their metabolic and respiratory processes, but also as a result of external stimuli including oxygen tension and light. The oxygen content in the oviduct and uterus is lower (2–8%) than that used in in vitro research (about 20%) [60,61]. In cows [62], sheep [62], goats [63], pigs [64], and mice [65], toxic impacts of atmospheric oxygen concentrations and beneficial impacts of lower ones on the outcome of IVEP were reported [66].

The normal development of oocytes occurs inside the female reproductive tract in complete darkness. On the other hand, they are rapidly exposed to light during their in vitro handling. This exposure has been shown to promote OS in embryos and somatic cells by either increasing pro-oxidant production or blocking endogenous antioxidant mechanisms [67]. One-cell hamster embryos exposed to 14,000 lux light for a brief time period (30 s) revealed a considerable rise in H_2_O_2_ levels [67]. 

The redox status of oocytes and embryos was found to be affected by the amounts of nutrients in the culture medium. Hashimoto, Minami [68] found that high glucose concentrations in the IVM medium hampered embryo development, which was accompanied with high ROS and low glutathione concentrations. On the other hand, bovine COCs exposed to 50–100 mM H_2_O_2_ for one hour during IVM showed some improvement in their subsequent development [69].

Several studies have been conducted on the role of cumulus cells in the redox status of COCs. Spermatozoa incubated with intact COCs produced more ROS than spermatozoa incubated with denuded oocytes [53]. The enhanced production of ROS in intact COCs was observed to improve the sperm penetration of oocyte membranes. Tatemoto, Sakurai [70] observed that after 44 h of culture under the hypoxanthine–xanthine oxidase system, denuded oocytes had more DNA and apoptotic damages than intact COCs. Furthermore, Luciano, Goudet [71] found that supplementing horse COCs with cysteamine boosted GPx mRNA expression in cumulus cells but not in oocytes in vivo and in vitro. Cetica, Pintos [72] also noted higher SODs, CAT, and GPx enzymatic levels in separated cumulus cells of bovine COCs during IVM compared to denuded oocytes, which decreased afterwards. In addition, removing cumulus cells before the in vitro maturation of bovine oocytes resulted in reductions in their maturation, fertilization and embryo development rates [73].

The role of reactive oxygen species (ROS) in the process of GVBD is controversial. Tamura, Takasaki [74] showed that high levels of H_2_O_2_ impaired GVBD during human oocyte maturation, which was reversed by melatonin. In swine oocytes, inhibiting SODs’ enzymatic activity resulted in a considerable decrease in meiotic development [75]. On the other hand, Takami, Preston [76] reported that the addition of a number of cell-permeable antioxidants to IVM medium for two hours blocked spontaneous GVBD in both intact COCs and denuded oocytes in rats. These antioxidants included nordihydroguaiaretic acid, 2(3)-tert-butyl-4-hydroxyanisole, octyl gallate, ethoxyquin, 2,6-di-tert-butyl-hydroxymethyl phenol, butylated hydroxytoluene, tert-butyl hydroquinone, propyl gallate, lauryl gallate, and 2,4,5-trihydroxybutrophenone. Tarin [77] also showed no variations in IVM rates in mouse oocytes when the oxidizing agent tertiary-butyl-hydroperoxide was added (tBH).

During oocyte development, a high risk of aneuploidy or chromosomal abnormalities has been reported, which becomes increasingly apparent during meiosis I [78]. Tarin [77] reported that OS induced by tBH treatment resulted in abnormal-shaped meiotic spindles associated with defects in the alignment of chromosomes. Tarin, Vendrell [79] also observed a modest reduction in oocytes’ aneuploidy from mice who received antioxidant supplementation.

Oocyte maturation is dependent on the proper assembly of the meiotic spindle. Several studies have related changes in the morphology of the meiotic spindle to gamete origin and culture circumstances during IVM [80,81,82]. During the IVM of mouse oocytes, Choi, Banerjee [83] reported that OS induced by H_2_O_2_ addition caused time-dependent changes in microtubule dynamics and chromosomal alignment. These changes were partially corrected by vitamin C supplementation via alleviating the harmful effect of H_2_O_2_ on chromosome alignment but not on microtubule alteration.

## 5. Antioxidant Supplementation during IVM of Mammalian Oocytes to Counteract ROS-Induced Damage

Currently, various antioxidant substances are being added during IVM to ensure a balanced intracellular redox status and good oocyte quality [84,85]. Adding antioxidants, such as thiols, polyphenolic compounds, melatonin, carotenoids, resveratrol, and vitamins such as C and E, to the IVM medium has been proven in several trials to improve oocyte quality and alleviate their damage induced by excessive ROS exposure [86,87]. Details of these antioxidants are listed in Table 1.

Thiols, or mercaptans, are organosulfur compounds similar to alcohols and phenols but have a sulfur atom in place of the oxygen atom [111]. Glutathione is a tripeptide thiol that naturally presents in either a reduced (GSH) or oxidized (GSSG) form [112]. It is made up of three amino acids: cysteine, glycine, and glutamate. It is a powerful reducing agent and also acts as a GPx electron donor. The presence of GSH in the oviductal fluid was reported [113]. Adding 0.6 mM cysteine, non-essential sulfur-containing amino acid, to bovine oocytes during IVM significantly increased embryo rates, regardless of exposure time [114]. 

Other thiols such as β-mercaptoethanol (β-ME) or cysteamine were added to boost intracellular GSH content and cell growth rates [115,116]. When these thiols were added to IVM medium in cows [117], sheep [118], pigs [119], dogs [120], and mice [98], growth-promoting effects were seen. Even at high oxygen concentrations, the addition of β-ME to the IVM medium was associated with increased embryo growth rates of mammalian embryos [119,121].

Polyphenolic compounds are plant metabolites that showed strong antioxidant activities [122]. Luteolin is a polyphenolic compound that revealed a major role in protecting several cell types against oxidative-stress-induced damage [123]. Luteolin improved the quality of porcine oocytes incubated under oxidative stress conditions and enhanced their subsequent embryonic development following IVF by alleviating oxidative damage to cell organelles [124].

Melatonin is a hormone that is produced naturally by the pineal gland. Melatonin receptors were found to be expressed in oocytes and cumulus cells of bovines [125,126] and mice [127]. The supplementation of IVM medium of bovine oocytes with exogenous melatonin significantly increased the rates of oocyte nuclear maturation, cleavage, and blastocyst formation [125,126]. Melatonin addition to the IVM medium also increased maturation and developmental competence in heat-stressed bovines [128] and pigs [129].

The supplementation of IVM medium with a mixture of 10 µM of alpha-tocopherol and 250 µM of l-ascorbic acid resulted in a larger proportion of denuded porcine oocytes progressing to the MII stage when compared with control groups [130]. A study by Choi, Banerjee [83] also suggested a beneficial role of ascorbic acid in protecting MII mouse oocytes from H_2_O_2_-induced damage.

Carotenoids are yellow- to red-colored naturally occurring pigments that are found mostly in plants, algae, marine organisms, and certain bacteria [131]. Over 1100 carotenoids have been reported [132]. They cannot be synthesized by mammals; thus, they must be obtained from the diet [133].

Carotenoids include α-carotene, β-carotene, and lycopene [134]. Xanthophylls are oxygenated derivatives of carotenes and represent the most abundant type of carotenoids [135]. Examples of xanthophylls are lutein, astaxanthin, and canthaxanthin. Carotenoids that contain one or more ketone groups are termed ketocarotenoids [136].

β-carotene supplementation to the IVM medium of mouse oocytes has been reported to reduce ROS levels, decrease cell apoptosis, and improve the overall structure of oocytes [137]. 

The supplementation of the IVM media of bovine and mouse oocytes with lycopene at a concentration of 0.2 μM resulted in a significant increase in the IVM rate compared to control oocytes, which was associated with reduced intracellular ROS levels and increased mitochondrial activities of oocytes [49,94].

Several keto-carotenoids have been employed in IVM experimental protocols. Canthaxanthin is a keto-carotenoid with significant antioxidant activity [138,139]. Canthaxanthin supplementation during IVM increased porcine oocyte maturation and subsequent developmental competence after parthenogenetic activation and somatic cell nuclear transfer [140]. 

Astaxanthin is another keto-carotenoid with potent antioxidant properties. Astaxanthin supplementation during IVM improved the maturation, fertilization, and development of pig oocytes [141]. 

## 6. Heat Stress and ROS Production by Mammalian Oocytes

### 6.1. Effect of Heat Stress on Oocyte ROS Production

Global warming is the steady rise in the Earth’s average temperature caused by increased rates of emissions of greenhouse gases, which trap heat and warm the Earth [142]. The rising global temperature exposes animals to stressful environmental circumstances, particularly in the summer, resulting in a reduction in domestic animal fertility due to an increase in body temperature exceeding physiological limitations, a condition known as heat stress [143].

Heat stress represents a potential risk for female infertility since it causes a series of physiological, metabolic, cellular, and molecular alterations in the reproductive tract [144]. Among the alterations induced by heat stress in different body tissues is the oxidative damage of intracellular components which results in structural and functional changes and causes apoptosis [145]. 

Heat stress has been reported to alter oocyte nuclear and cytoskeletal architecture and delay early embryonic development [146]. Heat stress stimulates excessive production of free radicals within maturing oocytes [38]. In cattle, the increased generation of free radicals within the oocyte following heat stress is associated with meiotic arrest, poor oocyte quality, and a lower rate of embryo development [147,148]. Moreover, the incubation of buffalo oocytes under heat stress condition led to an increase in the levels of ROS, lipid peroxide, and nitric oxide [149]. Additionally, mouse oocyte cytoplasmic maturation is more susceptible to thermal stress than nuclear maturation [150]. Furthermore, it was found that autophagy induction in pig oocytes is increased under heat stress conditions [151].

Several studies have shown that the process of oocyte maturation is disturbed at high temperatures [150,152,153,154]. The impact of elevated temperature on oocytes and embryos has been linked to a parallel increase in ROS production and a decrease in antioxidant defense enzyme activity [155,156,157,158]. In mouse oocytes and embryos, heat stress caused a decrease in the glutathione content and a rise in the H_2_O_2_ levels [156]. Culturing oocytes at high temperature during IVM has been also found to augment ROS production in bovines [159] and porcine [160]. 

It has been established that oocytes and embryos are the primary targets of the damaging effects of heat stress [161,162]. The cellular damage induced by heat stress involves diverse cellular organelles. Heat shock during the IVM of bovine oocytes has been reported to disturb spindle formation [163], mitochondrial function [164], microtubule and microfilament organization [165], and cortical granule distribution [166]. 

Breakdown of the germinal vesicle and expulsion of the first polar body are two significant characteristics of oocyte maturation [167]. Reduced rates of polar body extrusion and germinal vesicle breakdown were seen in oocytes matured at an elevated temperature (41.5 °C) for 22 h compared to oocytes matured at an optimum condition (38.5 °C) for the same time period in porcine [129,160], cattle [153,168] and buffalos [169]. On the other hand, heat stress (41 °C) increased the nuclear maturation kinetics of porcine oocytes, resulting in a larger proportion of MII oocytes after 16–18 h of IVM, according to Tseng, Tang [170].

### 6.2. Use of Antioxidants to Counteract ROS-Induced Damage of Heat-Stressed Oocytes

The possible involvement of antioxidant supplements in combating the negative effects of heat stress on oocyte developmental competence and IVEP is a subject of investigation [171]. The addition of retinol to the IVM medium of bovine oocytes prevented heat-induced reductions in oocyte maturation and improved the rate of blastocyst yield [172]. In mice, the administration of the antioxidant epigallocatechin gallate decreased the deleterious effects of maternal hyperthermia on follicle-enclosed oocytes via the suppression of ROS generation [173]. 

Porcine oocytes supplemented with 2 μM of resveratrol during their IVM under heat stress conditions revealed higher rates of polar body formation compared to heat-stressed oocytes with no resveratrol added [174]. 

The treatment of bovine COCs with astaxanthin at concentrations of 12.5 and 25 nM rescued the developmental competence of heat-shocked oocytes via enhancing SOD activity and decreasing ROS levels [175].

Melatonin has been shown to overcome the negative influences of heat stress on oocytes and preimplantation embryos [174,176,177,178]. Melatonin treatment upsurged SOD enzyme levels in heat-shocked mouse oocytes [156]. Melatonin addition at a 1 μM concentration decreased ROS levels of maturing bovine oocytes [179].

Coenzyme Q10 supplementation to IVM media improved mitochondrial properties and alleviated the effects of thermal stress on bovine oocytes’ developmental competence [180].

## 7. Maternal Aging and ROS Production by Mammalian Oocytes

### 7.1. Effect of Maternal Aging on Oocyte ROS Production

The aging of female mammals is widely established to be associated with decreased fertility [11]. The ovary is much more sensitive to the effects of aging than any other body tissue, and reproductive success has been shown to be inversely associated with age [181]. Ovarian senescence progressively reduces both follicle quantity and oocyte quality, resulting in a steady decrease in fertility which ends ultimately with sterility [182]. Mammalian oocytes display higher ROS levels with age, and thus, reproductive aging can be linked to the cumulative oxidative damage of oocytes [183,184,185].

Oxidative stress is one of the key processes underpinning aging. It arises as a result of the slow buildup of damage caused by free radicals created during normal metabolism [186]. The oocytes of most animals initiate meiosis in the fetal ovary. However, they enter a prolonged period of rest at the dictyate stage of the first meiotic prophase (prophase I) from fetal life until puberty. The duration of this period ranges from weeks in mice to months and years in domestic animals [182]. During this prolonged rest, physiological levels of generated ROS contribute to oocyte maturation within the follicle [187]. However, the cyclic production of these damaging agents negatively affects oocyte quality and may lead to an increased risk of ovarian pathologies, especially under circumstances of reduced antioxidant status [187]. 

Lipid metabolism is a powerful source of energy, and its relevance during oocyte maturation is becoming more apparent [188]. Lipid peroxidation is a hallmark of oxidative stress, which is known to increase as oocytes age [189]. Oocytes from aged mice were found to display significantly higher levels of lipid peroxides compared to those from adult mice [190].

### 7.2. Use of Antioxidants to Counteract ROS-Induced Damage of Maternally Aged Oocytes

In maternally aged oocytes, the disrupted balance between oxidant production and elimination leads to the development of oxidative stress [191]. The use of exogenous antioxidants is thus thought to antagonize the effects of maternal aging on oocytes via reducing oxidative stress. The oral administration of vitamins C and E reduced the negative effects of aging on ovarian reserve (number of both ovarian and ovulated oocytes) and oocyte quality (chromosomal aberrations and apoptotic changes) in a mouse model [192]. Melatonin and coenzyme Q10 exerted anti-aging effects on mouse oocytes through modulating mitochondrial activity and ROS levels during reproductive aging [193,194,195].

## 8. Postovulatory Aging and ROS Production by Mammalian Oocytes

### 8.1. Effect of Postovulatory Aging on Oocyte ROS Production

Ovulation is a complex process involving the rupture of the dominant ovarian follicle and the release of oocytes into the uterine tube [196]. The oocytes of domestic mammals are ovulated during the metaphase of the second meiotic (MII) division and stay in this state until fertilization [197]. Fertilization occurs when male and female gametes fuse within 10 h of ovulation in most domestic animals [198]; however, it can take up to 15 h in mice [199]. If an oocyte waits for a prolonged time without being fertilized by a viable sperm after ovulation, it will go through a series of deteriorating changes known as oocyte postovulatory aging [200].

The postovulatory aging of oocytes involves cellular and molecular alterations that reduce the developmental competence of oocytes [12]. The structural integrity of numerous oocyte components, including the zona pellucida, mitochondria, and meiotic spindle, has been shown to be adversely affected by postovulatory aging [201,202]. Furthermore, postovulatory aging has been linked to biochemical changes in oocytes, including an increase in the formation of ROS in mouse oocytes [203,204]. In postovulatory-aged mouse oocytes, a considerable increase in ROS levels has been linked to cell membrane disruption and DNA damage [205]. Increased ROS generation has been shown to lower intracellular ATP levels in aged bovine oocytes [206], decrease the glutathione (GSH)/glutathione disulfide (GSSG) ratio, and expedite oocyte fragmentation in mouse and pig oocytes [207,208]. 

Normally, there is a balance between the generation and removal of ROS. A growing body of research implies that the buildup of ROS in postovulatory-aged oocytes is progressively rising with time [209,210]. ROS accumulation is likely to be higher in in vitro matured oocytes than in in vivo matured ones due to laboratory conditions such as light exposure, high oxygen tension, and a lack of antioxidants derived from the surrounding ovarian and tubal microenvironments [8,211]. It was thought that the depletion of intracellular antioxidant defense system supplies in oocytes, such as glutathione, would exacerbate the oxidative damage caused by ROS buildup [212,213]. Consequently, incorporating antioxidant-rich compounds in programs used for both the in vivo and in vitro maturation of domestic mammal oocytes might reduce oxidative damage inflicted by postovulatory aging [202].

### 8.2. Use of Antioxidants to Counteract ROS-Induced Damage of Postovulatory Aged Oocytes

A number of studies have analyzed the effect of antioxidant supplementation on oocyte postovulatory aging in vitro. L-ascorbic acid and 6-methoxy-2,5,7,8 tetramethylchlormane-2-carboxylic acid (trolox/vitamin E) were shown to be ineffective in preventing oocyte fragmentation during postovulatory aging [214]. The reducing agent dithiothreitol (mda), on the other hand, increased oocyte quality, which resulted in higher rates of fertilization and blastocyst development [214]. Despite its protective effect on aging oocytes, the practical use of DTT is limited by its DNA damaging properties [215]. In vitro aged mouse oocytes treated with melatonin revealed a reduced rate of fragmented oocytes together with a decrease in ROS concentrations compared to counterparts aged without melatonin [210]. 

Mouse oocytes that were allowed to age in the presence of 200 nM lycopene showed significantly less fragmentation and degeneration, lower concentrations of H_2_O_2_ and MDA, and higher concentrations of TAC, GSH, and SOD than those aged without lycopene [216].

## 9. Conclusions

In vitro maturation (IVM) constitutes the basic foundation for several assisted reproductive techniques including IVEP. Subsequently, the successful practice of IVM holds great promise for the better conduction of these techniques. The disconnection of oocytes from their surrounding in vivo microenvironment during IVM makes them prone to very minor damaging insults. Additionally, the in vitro handling of oocytes during IVM exposes them to supraphysiological levels of stress that could result in their damage. Therefore, the success of an IVM protocol remains a great challenge to reproductive biologists.

The buildup of large amounts of ROS during IVM constitutes a startup point for oxidative stress that might increase the risk of IVEP failure. Among the several laboratory factors that contribute to ROS accumulation within maturing oocytes and IVM media are increased light intensities, high ambient temperatures, and the increased partial pressure of oxygen (PO2). Other factors include maternal senescence, oocyte aging caused by fertilization delays, and inadequate nutrients provided by IVM media (Figure 1). The latter factor is inevitable, as several components of the in vivo milieu cannot be simulated under standard laboratory conditions, e.g., blood circulation and signaling molecules. 

The lack of sufficient antioxidant protection for oocytes that is usually achieved by their natural intrafollicular habitat highlights the importance of antioxidant supplementation during IVM. Although the selection of antioxidant(s) to be added during the IVM of mammalian oocytes is a subject of rigorous investigation, generally, these substances are tested and applied at very small doses to achieve physiological levels similar to those found in mammalian biological fluids. As the follicular fluid represents the ideal site for oocyte development, future studies should be aimed at surveying and using species-specific follicular fluid resident factors to help in suppressing ROS activity during IVM. In addition to these factors, the testing of natural antioxidants with previously uncharacterized effects on oocytes and/or the combinatorial use of antioxidants with well-characterized effects on oocytes will definitely help to maximize the success rate of the IVM of mammalian oocytes.

## Figures and Tables

**Figure 1 vetsci-09-00439-f001:**
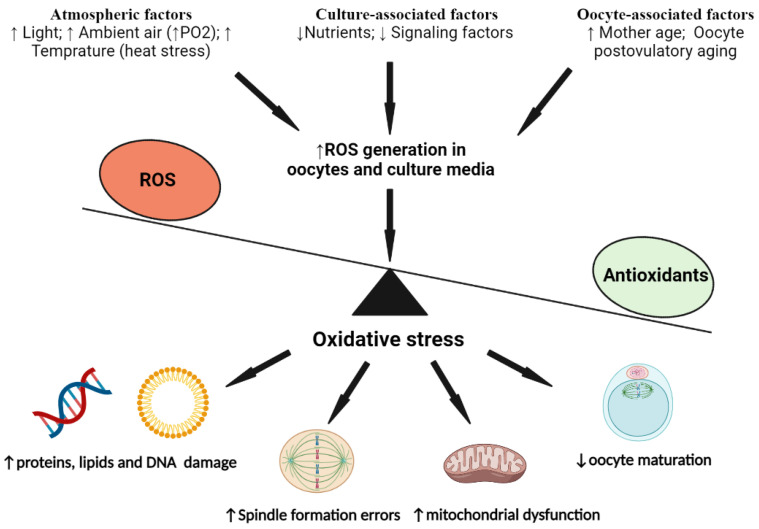
Schematic diagram summarizing the main factors causing oxidative stress of mammalian oocytes during their in vitro maturation. Consequences of oxidative stress on oocyte components are also shown. PO2, partial pressure of oxygen; ROS, reactive oxygen species. The diagram was created with BioRender.com [217].

**Table 1 vetsci-09-00439-t001:** List, concentrations, and effects of antioxidants used to improve in vitro maturation of oocytes in different animals.

Antioxidant	Type	Dose	Species	Maturation Rate vs. (Control)	References
Melatonin	Hormone	4.3 × 10^−8^ M (10 ng/mL)	porcine	84.6 (75.6) *	[88]
		10^−9^ M	bovine	82.3 (65.7) *	[89]
		2.5 × 10^−4^ M	buffalo	42.8 (33)ns	[90]
		10^−7^ M	sheep	85.3 (75.3) *	[91]
		10^−6^ M	mouse	85 (64) *	[92]
Lycopene	Carotenoid	2 × 10^−7^ M	bovine	76 (66.3) *	[93]
		2 × 10^−7^ M	mouse	89.9 (66.7) *	[94]
Astaxanthin	Carotenoid	2.5 × 10^−6^ M	porcine	89.5 (87.1)ns	[95]
Beta-Mercaptoethanol (β-ME)	Thiol	2 × 10^−5^ M	buffalo	76.2 (66.7)ns	[96]
		10^−5^ M	equine	55.6 (51.9)ns	[97]
Cystamine	Thiol	10^−5^ M	mouse	80.1 (57.7) *	[98]
Vitamin C	Vitamin	2.5 × 10^−4^ M	mouse	29.7 (70.3) *	[99]
		2.3 × 10^−3^ M (1 mg/mL)	bovine	~80 (~80)ns	[100]
Vitamin E	Vitamin	2.3 × 10^−3^ M (1 mg/mL)	bovine	~80 (~80)ns	[100]
		10^−3^ M	porcine	72.2 (67.6)ns	[101]
Selenium (SeMet)	Trace element	2.5 × 10^−8^ M	porcine	80.2 (67.6) *	[101]
Vitamin E; Selenium (SeMet)	Vitamin; trace element	10^−3^ M; 2.5 × 10^−8^ M	porcine	85.1 (67.6) *	[101]
Resveratrol	Polyphenolic compound	10^−6^ M	bovine	93.4 (87.9) *	[102]
		5 × 10^−6^ M	porcine	84.5 (72.6) *	[103]
Quercetin	Polyphenolic compound	10^−5^ M	mouse	86.6 (79.7) *	[104]
			human	92.3 (87.5)ns	
L-Carnitine	Amino acid derivative	3.1 × 10^−3^ M (0.5 mg/mL)	porcine	60.7 (56.4) *	[47]
		3.1 × 10^−3^ M (0.5 mg/mL)	camel	74.7 (60.2) *	[105]
		3.7 × 10^−3^ M (0.6 mg/mL)	canine	41.4 (23.4) *	[106]
Retinoic acid	Vitamin A metabolite	10^−8^ M	goat	78.7 (65.1) *	[107]
		2 × 10^−5^ M	camel	69.4 (52.9) *	[108]
Coenzyme Q10	Coenzyme	10^−5^ M	porcine	76.4 (66)ns	[109]
		5 × 10^−5^ M	human	82.6 (63.0) *	[110]

* Maturation rate significantly changed (*p* < 0.05); ns maturation rate non-significantly changed; SeMet, Seleon-L-methionine.

## Data Availability

Not applicable.

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
