# Peer review of "Importance of Antioxidant Supplementation during In Vitro Maturation of Mammalian Oocytes"

_vetsci, 2022, doi:10.3390/vetsci9080439_

Round 1
Reviewer 1 Report
Review Importance of antioxidants supplementation during in vitro 2 maturation of mammalian oocytes
This is a very through and current review about the benefits of using antioxidants to supplement mammalian oocytes during in vitro maturation. They emphasize the benefits of this supplementation to minimize the production of reactive oxygen species. The authors use the most current literature available and do a great job in including the most relevant factors for IVM in a comparative way.
Line 16: there is an additional space in “animal’s body”
Line 76: Rewrite as: the oocyte appears to contain a large nucleus,
Line 182: Elaborate more on the current research related to vitamin C in ROS and IVM.
Line 258: Please add effects of heat stress in other species along with the effects presented here for bovine.
I suggest adding the following research:
Zhang, L. I., et al. "Cumulus cell function during bovine oocyte maturation, fertilization, and embryo development in vitro." Molecular reproduction and development 40.3 (1995): 338-344.
The diagram at the end is a great way to integrate the discussed points.
Please, add a table to summarize common antioxidants used for IVM and their effects on the percentage of IVM of oocytes from in the literature.
Reviewer 2 Report
The authors have done a great job in compiling the updated literature on the effects of oxidative stress on the oocyte under maturation, and also the possible antioxidants used for this purpose.
The authors relied heavily on pre-existing literature for bovine and rodents, it would be interesting to know how other species behave.
If possible, I also suggest the inclusion, in topic 2, the literature that demonstrates evidence of uncoordinated maturation between oocyte´s nucleus and cytoplasm , often seen in horses, for example.
line 189: correct the word "excessive"
Both the conclusions and the figure presented are very well presented.
Reviewer 3 Report
This manuscript describes oxidative stress during IVM and potential protective agents to inhibit ROS production by oxidative stress. Although various existing reviews have been published, and novelty is somewhat lacking, a brief and definitive description of the multiple antioxidants used in IVM may intrigue readers. However, it is expected that the quality of the manuscript can be improved if minor revisions are made.
The author organized the main factors causing ROS production in Figure 1. If the author organizes the antioxidants used in IVM of oocytes in a “table” with references, readers can understand them more quickly and conveniently.
